# Bioactive Components of *Salvia* and Their Potential Antidiabetic Properties: A Review

**DOI:** 10.3390/molecules26103042

**Published:** 2021-05-20

**Authors:** Aswir Abd Rashed, Devi-Nair Gunasegavan Rathi

**Affiliations:** Nutrition, Metabolism and Cardiovascular Research Centre, Institute for Medical Research, National Institutes of Health, Ministry of Health, Malaysia, No.1, Jalan Setia Murni U13/52, Seksyen U13 Setia Alam, Shah Alam 40170, Malaysia; rathidevinair@moh.gov.my

**Keywords:** *Salvia*, bioactive components, antidiabetic, human health

## Abstract

The utilization of therapeutic plants is expanding around the globe, coupled with the tremendous expansion of alternative medicine and growing demand in health treatment. Plants are applied in pharmaceuticals to preserve and expand health—physically, mentally and as well as to treat particular health conditions and afflictions. There are more than 600 families of plants identified so far. Among the plants that are often studied for their health benefit include the genus of *Salvia* in the mint family, Lamiaceae. This review aims to determine the bioactive components of *Salvia* and their potential as antidiabetic agents. The search was conducted using three databases (PubMed, EMBASE and Scopus), and all relevant articles that are freely available in the English language were extracted within 10 years (2011–2021). *Salvia* spp. comprises many biologically active components that can be divided into monoterpenes, diterpenes, triterpenes, and phenolic components, but only a few of these have been studied in-depth for their health benefit claims. The most commonly studied bioactive component was salvianolic acids. Interestingly, *S. miltiorrhiza* is undoubtedly the most widely studied *Salvia* species in terms of its effectiveness as an antidiabetic agent. In conclusion, we hope that this review stimulates more studies on bioactive components from medicinal plants, not only on their potential as antidiabetic agents but also for other possible health benefits.

## 1. Introduction

According to World Health Organization’s 2019 Global Health Estimates, it was estimated that non-communicable diseases contribute to about 7 out of 10 causes of death globally. The prevalence of diabetes was noticed more rapidly in low and middle-income countries. Chronic diabetic complications are associated with various long-term complications that include prolonged injuries, organ failure, vision loss, renal impairment, peripheral neuropathy with foot ulcer risk and even amputation [1]. Currently, there are various therapies available to control diabetes, such as insulin, pharmaco and diet therapy. Despite the availability of multiple drugs that exert antidiabetic effects via various pathways, however, optimal treatment effects are yet to be achieved. In recent times, medicinal plant-based therapies have gained attention given their rich constituents (e.g., carotenoids, terpenoids, alkaloids, glycosides, flavonoids) that portray antidiabetic effects [2,3].

In the US, the development of herbal-based products as prescription drugs is subject to Food and Drug Administration approval. For example, Veregen^®^ (sinecatechins), a product derived from green tea (*Camellia sinensis*) intended for external genital or perianal warts, was first approved in the year 2008 [4]. In 2012, another product Crofelemer was introduced targeted for diarrhea relief among HIV/AIDS patients [5]. The extensive benefits of herbal plants have attracted much interest in their possible application-focused for health benefits, where it is usually commenced with chemical composition, in vitro and/or in vivo before any intervention studies. Chemical composition characterization assists researchers in ensuring the most efficient dosage that is required for evident changes of certain disease biomarkers. Moreover, further evaluations with in vitro- or in vivo-based approaches also strengthen the findings in determining the most appropriate extract concentration without any adverse effects.

There are plenty of medicinal plants worldwide, and among these, *Salvia* spp. was acknowledged for its excellent medicinal benefits. The genus *Salvia* L. belongs to the Lamiaceae family and shows about 900 species dispersed worldwide, mainly in the areas of the Mediterranean, Southeast Africa, Central and South America [6]. The *Salvia* name comes from the Latin word “salvare” which means “to heal” and “to save”. *Salvia* spp. have been reported to produce various phenolic metabolites that have gained much attention in relevance to their antioxidant, antimicrobial, antidiabetic, neuroprotective, anti-inflammatory as well as cytotoxic properties [7,8]. Earlier studies have revealed the crucial role of phenolic components and flavonoids in both α-amylase and α-glucosidase inhibition [9,10]. These are classified as the key enzymes that catalyze carbohydrates into glucose monomers where the presence of inhibitory components assists in regulating optimal glucose levels among diabetes patients [11]. One of the most commonly used species of sage is *S. miltiorrhiza*. *S. miltiorrhiza* is a perennial flowering plant that grows along stream banks, hillsides and grassy forests that is elevated at 90–1200 m. The roots of *S. miltiorrhiza Bunge,* also called Danshen in Chinese, are one of the famous parts of the plant, which is common to Eastern Asia and included as an official drug in the 2010s Chinese Pharmacopeia [12]. Another commonly used species is *S. officinalis* that also has been used as a traditional remedy against diabetes in many countries for its glucose-lowering effect. The evaluation of *S. officinalis* tea infusion showed comparable efficacy with metformin, which is normally used for type II diabetes treatment [13]. Diabetes mellitus (DM) is a complex chronic illness that occurs due to insulin secretion deficiencies associated with high blood glucose levels. Type 2 diabetes mellitus (T2DM) is the most common and prevalent disease that is of major public health concern [14]. T2DM is a progressive disease characterized by a slow and continuous decline in β-cell function [15]. α-Glucosidase and α-amylase are the enzymes involved in carbohydrate digestion, where their inhibition significantly reduces the postprandial increase of blood glucose. Hence, this serves as a strategic approach in blood glucose management among T2DM and borderline patients [16].

Hydrolysis of starch is possible by enzymes from amylases (endoamylases and exoamylases). The most well-known endoamylase is α-amylase, a calcium metalloenzyme that cannot function in the absence of calcium. α-amylase is present in both the salivary and pancreatic amylases of humans and certain plants, fungi and bacteria. α-amylase inhibitors are also known as starch blockers that exert their antidiabetic mechanism by preventing starch absorption through blocking hydrolysis of 1,4-glycosidic linkages of starch and other oligosaccharides into maltose, maltotriose and other simple sugars [17]. The hydrolysis degree of the substrate by α-amylases can be divided into two categories, where saccharifying α-amylases hydrolyze 50–60% and liquefying α-amylases cleave about 30–40% of the glycosidic linkages of starch.

α-Glucosidase is an essential enzyme in lysosomes for glucose degradation [18]. The enzyme inhibitors are capable of binding reversibly to the carbohydrate-binding region of α-glucosidases. This bonding leads to competition with oligosaccharide binding and thus, delays oligosaccharides’ cleavage into monosaccharides. These mechanisms highlight the action of α-glucosidase in retarding intestinal glucose absorption and postprandial hyperglycemia (PPHG) [19]. PPHG is one of the risk factors in diabetic patients that were identified to complicate T2DM treatment. Indirectly, α-glucosidase inhibitors play an important role in treating PPHG in diabetic patients [20]. Thus, the hydrolysis of oligosaccharides, trisaccharides, and disaccharides by membrane-bound intestinal α-glucosidase into glucose and other monosaccharides in the small intestine will be reduced, resulting in lower glucose circulation in the blood.

Acarbose, which is classified as an α-glucosidase inhibitor, is usually applied as a positive control in antidiabetic research. Acarbose plays its role by lowering carbohydrate digestion rate and slows down carbohydrates absorption from the digestive tract. This mechanism assists in lowering after-meal glucose effects that suggest their potential in the prevention of diabetes occurrence [21]. Acarbose also has been reported as a competitive inhibitor of α-amylase, where they are composed of the pseudo-sugar ring and glycosidic nitrogen linkage that mimics transition state for enzymatic cleavage of glycosidic bond and thus, inhibits α-amylase competitively [22]. The mechanisms of starch hydrolysis into glucose via the action of α-amylase and α-glucosidase are presented in Figure 1.

In vivo studies will require experimental animals to be induced as a diabetic model. Alloxan and streptozotocin (STZ) are the most common diabetogenic agents used to assess antidiabetic and hypoglycemic components. These materials inflate and eventually degenerate β-cells from the Langerhans islets [24]. Injection of the anterior hypophysis extract is a less effective approach for developing diabetes [25]. The final signs of insulin deficiency are seen in STZ-chemically impaired rats [26]. An autoimmune process that destroys the β-cells of the Langerhans islets will begin once a 60 mg/kg STZ dose was given to the rats. The use of 60 mg/kg STZ dose resulted in the toxicity of β-cells with the emergence of clinical diabetes within 2–4 days [27]. Alloxan is lower in cost and more readily available than STZ. On this basis, in experimental diabetes research, one would logically assume a preference for using alloxan. Chemically known as 5,5-dihydroxyl pyrimidine-2,4,6-trione, alloxan is an organic compound, a derivative of urea, a carcinogen and an analog of cytotoxic glucose [28]. Nevertheless, alloxan has poor diabetogenicity and is very normal with intraperitoneal (IP) doses of 150 mg/kg and below [29,30] for fast auto-reversal of alloxan-induced hyperglycemia.

In this review, we intend to conduct evidence-based evaluations on the potential of *Salvia* spp. as an antidiabetic agent focused on their chemical constituents, in vitro, in vivo and intervention-based studies.

## 2. Materials and Methods

### Search Strategy

The original articles were identified through searches of three databases (PubMed, Embase and Scopus) from 2011 to 2021 using the medical subject heading (MeSH) terms “*Salvia*”, crossed with the term “diabetes”. Publications with available abstracts were reviewed and limited to studies published in the English and Malay languages. Papers on chemical composition, in vitro, in vivo and human studies, and related to diabetes were included. However, review articles and letters to the editor were excluded. Duplicate articles were eliminated.

## 3. Results

After conducting a comprehensive literature review, the articles were selected and divided into several categories. A total of 58 articles were included in our first screening that consists of 6 articles on a combination of chemical composition and in vitro studies, 45 articles on a combination of chemical composition and in vivo studies, and 8 intervention studies. Upon second screening to include only articles with details of chemical components, only 28 articles were included as presented in Table 1. Among these, 6 articles were focused on a combination of chemical composition and in vitro studies, 18 articles on a combination of chemical composition and in vivo studies, and 4 articles on intervention studies. All the related articles were printed out for further evidence-based assessment to explore the effectiveness of *Salvia* spp. as a potential antidiabetic agent.

## 4. Discussion

### 4.1. Chemical Composition of Salvia spp. and Their Antidiabetic Effect via In Vitro and In Vivo Studies

Chemical composition analysis is acknowledged as the main focus, especially in studies investigating their potential benefits in the prevention or treatment of diseases. Normally, drug candidates will be examined by in vitro, in vivo and preclinical investigations to prove their safety and efficacy before they can be tested in humans. However, chemical constituents remain as the root that exerts their specific effects. Based on our findings, *S. virgata, S. viridis, S. urmiensis, S. syriaca and S. nemorosa* were some of the common species discussed with the presence of polyphenols, flavonoids and terpenoids identified as the major components [11,31,32,33,34]. Rosmarinic acid (RA), salvianolic acids, ursolic acid, oleanolic acid, chlorogenic acid, 1,8-cineole, thujone, β-pinene, spathulenol, linalool, linalyl acetate, rutin and caryophyllene oxide were among the main chemical constituents characterized based on the included studies from the various *Salvia* spp. The chemical structures of several major constituents are presented in Figure 2.

Chemical characterization was mainly performed using high-performance liquid chromatography (HPLC) and gas chromatography (GC) techniques equipped with various relevant detection systems, such as diode array detector (DAD), mass spectrometry (MS) and flame ionization detector (FID) [11,32,33,34]. Also, total phenolic content and total flavonoid contents were analyzed by the commonly applied Folin–Ciocâlteu and colorimetric techniques [33,34]. Research on *S. virgata* involved a fractionation study that involved various other techniques, such as column chromatography, preparative layer chromatography and nuclear magnetic resonance (NMR) [31]. In general, almost all included articles focusing on chemical profiling were followed by either in vitro or in vivo investigations. A study by Nickavar and Abolhasani in 2013 [31] reported that the ethanolic extract of *S. virgata* showed a dose-dependent α-amylase inhibition with IC50 of 19.08 mg/mL. One significant finding of this study is on the isolation and identification of active flavonoid chrysoeriol that also inhibited α-amylase activity (IC50 = 1.21–1.33 Mm). It was documented that the presence of a -OMe group at 3′- position of B-ring improves the ability to inhibit α-amylase activity [69]. In 2019, Zengin and colleagues [32] evaluated phytochemical composition and enzyme inhibitory potential of *S. viridis* ethanolic root extracts that were obtained by various methods. The choice of extraction technique was seen to influence phenolic and flavonoid composition, where UAE gave the highest concentration. UAE technique is based on applying high-frequency sounds and a limited amount of solvent to produce an effective extraction of the components contained in a solid matrix [70]. Among all extracts, the main components identified were mostly comprised of salvianolic acids, polyphenols, flavonoids and terpenoids. The enzyme inhibition potential of *S. viridis* was higher against α-glucosidase (1.61–1.65 mmol ACAE/g) compared to α-amylase (0.56–0.73 mmol ACAE/g), which presented a possible therapeutic approach for diabetes management [32].

An antidiabetic potential of essential oils (EOs) from *Salvia* sp. has also been investigated in several studies. Bahadori and team [11] evaluated EO composition and antidiabetic properties of *S. urmiensis*. EO analysis indicated a high presence of ester compounds in leaves (ethyl linoleate, methyl hexdecanoate and methyl linoleate), while 6,10,14-trimethyl-2-pentadecanonen was the major identified compound in flowers. Enzyme inhibition assays performed with EO and various extracts showed that methanolic extract gave the highest α-glucosidase and α-amylase inhibition with the lowest IC50 values (IC50 = 8.3 and 24 μg/mL). *S. urmiensis* EO was classified as a weak inhibitor that could be correlated with the presence of esters, ketone and alkane compounds. Similarly, another study attempted to investigate the phytochemical composition of *S. syriaca* EO and methanolic extract as well as antidiabetic properties. Spathulenol, isospathulenol and bornyl acetate were the major identified compounds in EO, while rutin, quercetin, apigenin, RA, and ferulic acid were the most abundant phenolic compounds. Contradictory to *S. urmiensis*, EO of *S. syriaca* exhibited the strongest activity in both α-glucosidase (IC50 = 1.18 mg/mL) and α-amylase assays (IC50 = 1.54 mg/mL) compared to all other tested extracts [33]. In the year 2017, Bahadori et al. [34] studied the chemical composition of *S. nemorosa* EO and phenolic compounds of methanolic extract along with α-glucosidase inhibition assay. The results demonstrated the strongest inhibition assay for the methanolic extract. RA was detected as the major compound of extract, whereas oxygenated sesquiterpenes constitute the highest proportion in EO. It was reported that the presence of RA correlates proportionally to α-amylase inhibition, where higher inhibition is observed with increased RA concentration [71].

Two structurally isomeric pentacyclic triterpenes compounds (ursolic acid and oleanolic acid) isolated from many *Salvia* spp. also have gained massive attention in relevance to their potent inhibition effects on α-glucosidase activity [72,73]. However, the analysis of these compounds encountered difficulties due to their structural similarities. A study by Janicsak et al. [74] presented the largest data of these acids among *Salvia* sp. in Turkey, although the study was restrained in terms of its quantitative evaluation. To counteract this gap, a group of researchers in 2018 embarked on investigating and developing techniques for the simultaneous analysis of both acids among fourteen *Salvia* spp. (*S. adenocaulon, S. aucheri var. aucheri, S. blepharochlaena, S. cilicica, S. absconditiflora, S. divaricate, S. euphratica var. euphratica, S. heldreichiana, S. huberi, S. hypargeia, S. limbate, S. rosifolia, S. sclarea* and *S. virgata*), along with their α-glucosidase enzyme inhibitory effects. The finding showed that *S. aucheri var. aucheri* and *S. adenocaulon* were the two species with the best α-glucosidase activities (IC50 = 17.6 and 25.9 µg/mL), respectively. Moreover, a strong negative correlation was reported for both ursolic and oleanolic acid with IC50 results of r = −0.623 and r = −0.695, respectively. Based on the collective findings, the authors concluded that α-glucosidase inhibitory activity could be attributed to the presence of both triterpenoids, suggesting their potential use as antidiabetic agents [35].

In addition to in vitro evaluation, potentials of *Salvia* spp. were also investigated through in vivo approaches using several species (*S. miltiorrhiza, S. fruticosa, S. sclarea*) as well as specific compounds, such as salvianolic acids, tanshinone IIA (TSIIA) and RA. Studies involving these specific compounds are mostly conducted using those that are commercially available. On the other hand, extract from respective plants varied from water extract, ethanol extract and EOs. The doses were given through intraperitoneal injection from 3 up to 24 weeks, and some were administered orally [38,39,42,44,47].

Salvianolic acids, as water-soluble compounds, are abundantly present in *S. miltiorrhiza,* with more than 10 different forms. Among these, salvianolic acid A (SalA) and B (SalB) are recognized as the most abundant compound [75]. Danshensu (DSS) ((*R*)-3-(3, 4-dihydroxyphenyl)-2-hydroxypropanoic acid) is the basic chemical structure that forms the various salvianolic acids [76,77]. Specifically, SalA is formed by a combination of DSS with two molecules of caffeic acid, whereas three molecules of DSS and one molecule of caffeic acid forms the SalB [75,78]. Previous researchers demonstrated extensive pharmacological effects of SalA that include antioxidative, anti-platelet aggregation, anti-cerebral, myocardial ischemia, and ameliorates diabetes complications [36,79,80,81]. On the other hand, SalB showed excellent protection against high-fat-diet-induced obesity, protects β-cells against cytotoxicity, prevents high glucose-induced apoptosis and also exerts hepatoprotective effects [82,83,84,85]. In a study conducted by Qiang et al. [38], the antidiabetic effect of SalA was indicated by the improvement of mitochondrial roles and stimulating AMP-activated protein kinase (AMPK) via the CaMKKβ/AMPK signaling pathway. This study proposed that SalA could potentially be applied for diabetes treatment in relevance to its appealing effects by reducing mitochondrial membrane potential (MMP) while increasing adenosine triphosphate (ATP) synthesis.

Based on the study by Yu and colleagues [37], the beneficial effect of SalA on peripheral nerve function was reported in diabetic rats. This result might be attributed to improvements in glucose metabolism by regulating the AMPK-proliferator-activated reaction-α-sirtuin 3 (PGC1α-Sirt3) axis. AMPK functions as a fuel sensor in several tissues, including skeletal muscle [86]. AMPK enhances peroxisome expression in the mRNA AMPK activation, PGC1α and manganese superoxide dismutase (MnSOD) [87]. In addition, PGC-1α serves as an important transcriptional coactivator for Sirt3 expression [88] and a pivotal factor for mitochondrial function [86]. As a member of the sirtuin family, Sirt3 regulation of silent matting-type information is located in the mitochondria and controls several pivotal pathways via targeted central metabolism enzymes [89].

SalA was found to lower fasting blood glucose (FBG) and fed blood glucose in a dose-dependent manner, as well as reduced 24-h food and water intake in a study conducted by Qiang et al. [38] using alloxan-induced type 1 diabetic mice with an HFD and low-dose STZ-induced T2DM rats. By using the HepG2 cells and L6 myotubes, SalA caused a dose-dependent increase in glucose consumption and enhanced glucose uptake. Moreover, SalA also decreased mitochondrial function, increased ATP production and decreased MMP via the CaMKKβ/AMPK signaling pathway. Intriguingly, SalA did not show any effect on insulin secretagogue and activation of PI3K/Akt signaling pathway. The pathway PI3K/AKT regulates the proliferation, differentiation and transformation of cells, as well as the metabolism and cytoskeletonization, which result in apoptosis and the survival of cancer cells. Therefore, numerous disorders, such as obesity, diabetes and cancer, are associated with this pathway. Thus, SalA has this unique function as it does not activate phosphorylation of Akt based on the recorded result by Western blot analysis.

SalB is also recognized for its potential use as antidiabetic agent where it resulted in significant reduction of serum glucose (*p* < 0.05–0.01) and MDA (*p* < 0.05), while serum insulin, glutathione (GSH) (*p* < 0.05) and catalase activity (*p* < 0.01) increased upon continuous administration of 20 or 40 mg/kg (IP injection) up to 3 weeks with no evident changes on nitrite of STZ-induced diabetic rats [39]. Raoufi and colleagues [39] presented the mechanism of the antidiabetic effects of SalB evaluated with multiple low-dose STZ-induced diabetes models where it was concluded that the action could be via protection of pancreatic β-cells against chemicals followed by improvement of the β-cells insulin secretion. Normally, SalB accounts for 3–5% of the total herbs dry weight [90]. Oral treatment of SalB at a much higher concentration (50 and 100 mg/kg) in a spontaneous model of T2DM mice (db/db) was found to decrease FBG, TG and free fatty acid levels, reduced hepatic gluconeogenic gene expression and improved insulin intolerance. Huang and his friends also found that high dose SalB significantly improved glucose intolerance, increased hepatic glycolytic gene expression and muscle glycogen content, and ameliorated histopathological alterations of the pancreas, similar to metformin [40]. In contrast, serum insulin was found to be higher in this study than the other study done by Raoufi and his colleagues [39].

Although many researchers focused on SalA for antidiabetic study, SalB was thought to have much more commercial value for food and medicine purposes due to the containment of the highest amounts in *S. miltiorrhiza* [91,92]. In another study that used a much higher concentration of SalB (50, 100, and 200 mg/kg), a significant decrease of blood glucose and insulin, along with increased ISI, was noticed at 100 and 200 mg/kg. They also found a significant decrease in TC, LDL, non-esterified fatty acids, hepatic glycogen, and muscle glycogen, and increased HDL, which was originally altered by HFD and STZ. Moreover, SalB (200 mg/kg) markedly decreased TG and MDA and increased superoxide dismutase, which was originally altered by HFD and STZ [41].

As previously mentioned, RA is also a chemical constituent in *Salvia* sp. RA can be found particularly in Lamiaceae and Boraginaceae family [93], and it was detected as one major component among two species, namely *S. nemorosa* and *S. syriaca*. RA occurs in nature as a phenolic compound. It is an ester of caffeic acid and 3,4-dihydroxy phenyl lactic acid with noteworthy biological roles, such as antioxidant, antidiabetic, anti-inflammatory, cardioprotective, hepatoprotective, nephroprotective and many others [94]. RA was shown to be metabolized in the intestines and liver upon absorption. Metabolites, such as caffeic acid, ferulic acid and coumaric acid, were noticed to be present along with RA upon its administration to rats. A study conducted by Azevedo and colleagues [42] investigated the effects of SFT treatment and RA as its major phenolic constituent against SGLT1, the facilitative GLUT2 and GLP-1. It was reported that without significant changes in total levels of cell transport proteins, RA from SFT was capable of increasing SGLT1. Nevertheless, there was no impact on GLUT 2, Na^+^/K^+^ -ATPase or GLP-1 levels by SFT. The findings of this study showed that SFT and RA specifically regulate SGLT1 transporter levels at the brush border membranes (BBM). SGLT1 transporter levels are controlled by decreasing their level in diabetic conditions and with an increased presence of digestible carbohydrates. This phenomenon is also in parallel with lower blood glucose where RA was recognized as the active component [42].

1,8-cineole, also known as eucalyptol, is a bicyclic monoterpene that can be obtained from various plant EOs, including *Salvia* sp. [95,96]. 1,8-cineole has been reported for its extensive pharmacological benefits, such as antimicrobial, anti-inflammatory and pain relief. A study by Kim and colleagues [43] investigated the potentials of eucalyptol in opposing diabetic kidney disease characteristics represented by renal tubular epithelial derangement and tubulointerstitial fibrosis. The findings of this study presented eucalyptol as a potent inhibitor of Snail1 and β-catenin in diabetic models of renal tubular cells and kidneys. Linalool is a volatile flavor and exists in two forms known as *R* (−)-linalool (licareol) or *S* (π)-linalool (coriandrol) that varies according to climate conditions. This compound has been acknowledged for strong antidiabetic characteristics based on previous investigations. Raafat and Habib [44] attempted to study the antidiabetic effect of *S. sclarea* EO from two different regions in Lebanon; Beirut (SS-Bt) and Taanayel (SS-TI). The phytochemical analysis found that SS-Bt was characterized by high linalool concentration, while linalyl acetate constitutes the SS-TI. This research monitored the acute and subchronic antidiabetic properties of EOs along with linalool and linalyl acetate, where the most active chemotype was determined. Results of this study indicate better activities with chemotype 1 that is characterized by higher linalool content.

Other compounds that can be found in *Salvia* spp. include lithospermic acid, cryptotanshinone, and TSIIA [45,46,47,48]. It was found that the treatment with LAB prevented vascular leakage and basement membrane thickening in retinal capillaries in diabetic rats, which indirectly thwart developing diabetic retinopathy in this animal [45]. On the other hand, cryptotanshinone did not affect FBG levels in STZ-induced rats. However, its action is more pronounce in inhibiting developing fibrosis to improve cardiac function by reducing the mRNA and protein levels of the STAT3, matrix metalloproteinase-9, and connective tissue growth factor in Type 1-like diabetic rats as well as in high glucose-cultured cardiomyocytes [46]. While TSIIA was found to display inhibitory effects on renin activity of HEK-293 cells; moreover, it downregulated protein expression of ANG II in human renin-expressed HEK-293 cells [47]. The determination of plasma renin activity has been widely used to evaluate the renin-ANG system in disease states. Therefore, measurement of plasma renin activity has been suggested as an important aid in the differential diagnosis of primary and secondary aldosteronism [97,98]. A differentiation between low and high renin hypertensive states may also help select antihypertensive as well as antidiabetic medications. Based on the animal study, treatment of diabetic mice with TSIIA at two different doses (10 and 30 mg/kg) found to decrease the significant level of ANG II in serum (from 16.56 ± 1.70 to 10.86 ± 0.68 and 9.14 ± 1.31 pg/mL) and managed to reduce the expression of ANG II in bone, consequently improving trabecular bone mineral density and microstructure of proximal tibial end and increasing trabecular bone area of distal femoral end in diabetic mice [47]. The potential antidiabetic effect is not only performed by the individual compound in *Salvia*. The total PAF of *S. miltiorrhiza Bunge* that contains SalA, SalB, RA, cryptotanshinone, TSIIA and other polyphenolic acids were also found to induce a significant decrease in FBG, FINS, TC, TG and BUN, along with an obvious ISI increase in STZ-induced diabetic rats [49].

Combined supplementation of Pae with DSS prevented vascular damage and improved vascular reactivity in STZ-induced diabetic rats. Moreover, phenylephrine (PE)-induced contraction response was also decreased in the treated groups. The combination showed significant protective effects by reducing oxidative stress and intracellular Ca^2+^ regulatory mechanisms [50]. Another study attempted to evaluate the effect of DSS on cognitive decline and AGE-mediated neuroinflammation in learning and memory deficits of STZ-induced diabetic mice. The outcome of this study presented the potential of DSS in ameliorating cognitive decline via attenuation of AGE-mediated neuroinflammation, which suggests its application as an alternative in preventing diabetes-associated cognitive impairment [51].

Jiangtang decoction (JTD), a patented drug that contains *Euphorbia humifusa Willd, S. miltiorrhiza Bunge, Astragalus mongholicus Bunge, Anemarrhena asphodeloides Bunge and Coptis chinensis Franch*, has also been studied for its antidiabetic properties. JTD exhibited a significant amelioration in glucose and lipid metabolism dysfunction, reduced morphological changes in the renal tissue, decreased urinary albumin excretion, and normalized creatinine clearance. JTD therapy decreased AGE, and RAGE accumulation improved IRS-1 and increased both PI3K (p85) and Akt phosphorylation suggesting the involvement in the PI3K/Akt pathway activation. JTD administration further decreased the high rates of renal inflammatory mediators and reduced NF-κB p65 phosphorylation [52]. The schematic pathway showing developing diabetic nephropathy, cardiomyopathy and nephropathy through the modulation of nitric oxide, protein kinase G (PKG-1), thrombospondin 1 (TSP1), transforming growth factor-β1 (TGF- β1) and NF-κB is shown in Figure 3.

It is known by the scientific community that diabetes is linked to various metabolic deranges, such as overproduction of reactive oxygen, hypoxic conditions, mitochondrial dysfunction and inflammation [88]. Poor diabetes control could damage blood vessel clusters in the kidney, resulting in kidney damage and elevated blood pressure. Prolonged stress conditions induce further strain to the kidney’s filtration system that could completely damage the kidney function. Orally given decoction from HDD significantly decreases excretion of urinary albumin and improvement in db/db mice at a dose of 6.8 g/kg/day for 12 weeks. HDD is composed of *Astragali Radix* (Huang-qi) and *S. miltiorrhizae Radix et Rhizoma*. The study by Liu and colleagues [53] reported that HDD treatment reversed the effects of enhanced mitochondrial fission and PINK1/Parkin-mediated mitophagy in the db/db mice. This finding suggested that administration of HDD tends to suppress PINK1/Parkin mediated mitophagy, and this mechanism plays a vital role in the protection of possible kidney injuries. The representation of PINK1/Parkin-mediated mitophagy is presented in Figure 4.

### 4.2. Salvia spp. in Intervention Studies

Four intervention studies were included in this review. On the whole, several *Salvia* spp. were investigated for their potential as antidiabetic agents. *S. miltiorrhiza* is the most studied, with three articles focusing on this species. In a study by Lian et al. [54], the safety and effectiveness of danshen-containing Chinese herbal medicine intended for diabetic retinopathy (DR) were conducted via randomized, double-blind, placebo-controlled multicenter clinical trial. This research involves a dose-ranging study where varying doses of CDDP were given for selected nonproliferative DR patients. The findings indicated that the observed effects were significantly better with those of high-dose and mid-dose treatment groups comparatively to low-dose treatment groups. However, the researchers stressed the study limitation whereby short clinical duration may not be sufficient enough that raises the need for future trials to evaluate if the extended duration also exerts a similar effect. Nevertheless, the positive outcomes observed the proposed potential role of Chinese herbal medicines in NPDR treatment and delaying possible progression [54].

In the year 2019, another study by Tassadaq and Wahid [55] attempted to evaluate the efficiency of Tricardin (danshenform 250 mg dripping pills capsules) along with physical rehabilitation among diabetic patients with polyneuropathies. Diabetic polyneuropathy was recognized as one of the major complications that induce pain, numbness or loss of sensation, which could ultimately result in amputation [102,103]. Apart from good glycemic control, proper use of footwear, as well as physical activities, are deemed vital in diabetes management, especially neuropathies [104]. This study highlighted that the addition of Tricardin pills to their usual treatment regime significantly led to an improved quality of life post-treatment, compared to the control group that only practiced conventional therapy. Hence, Tricardin is considered a safe and alternative option in the management of diabetic polyneuropathies, although long-term efficacy studies should be performed with a larger population [55].

In contrast to danshen pills, the effect of *S. miltiorrhiza* injection combined with telmisartan in DN was studied in 2018 [56]. This case–control study showed that increased levels of ColIV and FN are partially accountable for the progress of DN. This induction could be successfully altered via *S. miltiorrhiza* injection with telmisartan that attenuates both ColIV and FN, postpones the ultrastructure changes of the glomerular basement membrane and reverses the hyperglycemia state along with low adverse reactions. Nevertheless, future studies are suggested to be conducted longer with larger sample size and dose and time-dependent based approaches [56].

*S. officinalis* is another species of *Salvia* that also has been widely investigated with hypoglycemic activity. *S. officinalis* is comprised of volatile oils, tannins, diterpenes, triterpenes, steroids, flavones and flavonoids [13,105,106]. They are well-known for various medicinal benefits (antioxidant, anti-inflammatory, anti-hyperglycemic, anti-dyslipidemia as well as culinary uses [107,108,109,110,111,112]. A study by Kianbakht et al. [57] investigated the effects of *S. officinalis* leaf extract in combination with statin therapy in hypercholesterolemic T2DM patients, with positive findings that were deemed safe and further improved lipid profiles. These effects observed with glycemic control and lipid profile are portrayed as beneficial in preventing cardiovascular-associated complications among the patients [57].

## 5. Conclusions

The significance of *Salvia* spp. potentials in diabetes management were noticed based on the included studies ranging from chemical composition analysis, in vitro, in vivo, and clinical investigation. Collectively, *Salvia* spp. has gained much recognition, especially in China, and it is anticipated that more investigations are performed to establish their potential usage in all parts of the world. However, we would like to state the limitation of this review where some studies were conducted with commercially available compounds that we believe may differ in certain purity compared to the original extract. The mechanism of action for specific compounds remained elusive, and therefore, we cannot conclude the most effective *Salvia* spp. in diabetic management from our perspective. Despite this challenge, *Salvia* spp. have shown the huge potential to be explored as an alternative strategy in diabetic therapies, and we hope that this review will instigate more interest among researchers around the globe to further explore bioactive components from *Salvia* spp., as well as other medicinal plants for their potential not only as antidiabetic agents but also for other possible health benefits.

## Figures and Tables

**Figure 1 molecules-26-03042-f001:**
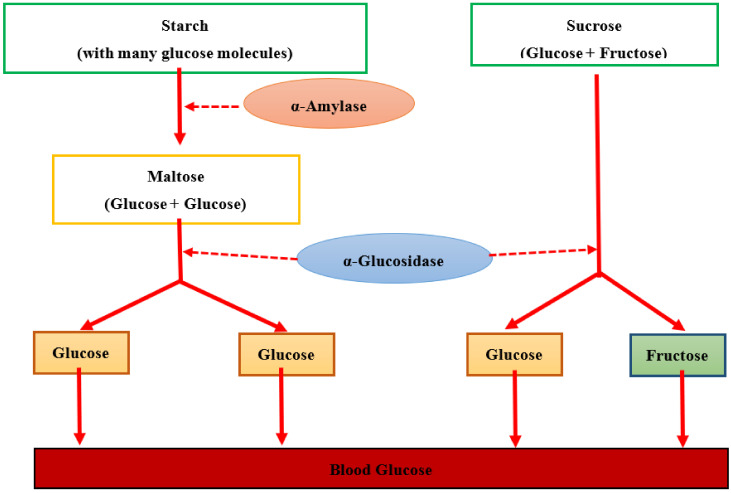
Hydrolysis of starch to glucose as catalyzed by α-amylase and α-glucosidase (adapted from [23]).

**Figure 2 molecules-26-03042-f002:**
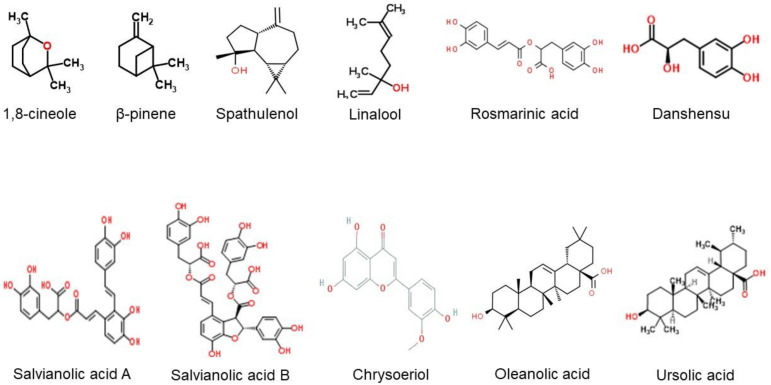
The chemical structures of several main chemical constituents (source: [58,59,60,61,62,63,64,65,66,67,68]).

**Figure 3 molecules-26-03042-f003:**
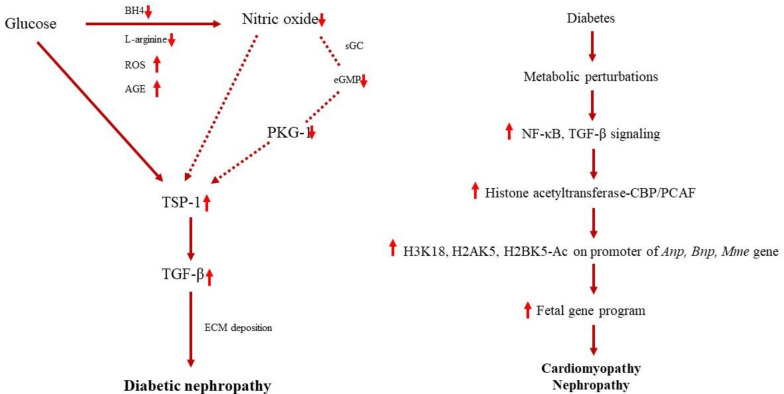
Schematic diagram representing the development of diabetic nephropathy, cardiomyopathy and nephropathy (adapted from [99,100]).

**Figure 4 molecules-26-03042-f004:**
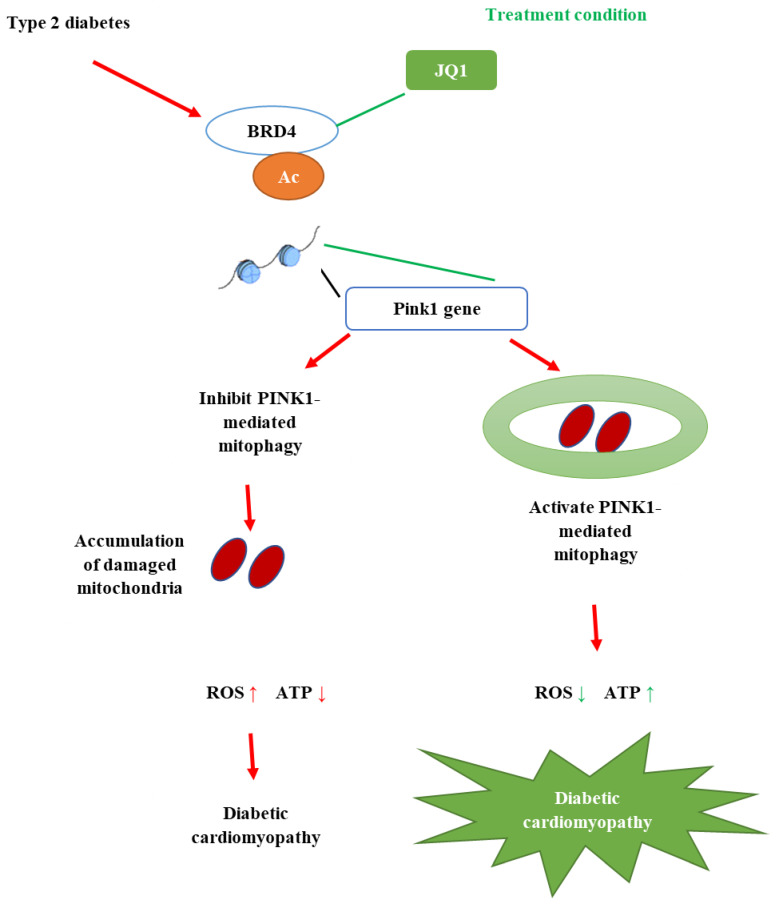
The schematic diagram of PINK1/Parkin-mediated mitophagy (adapted from [101]).

**Table 1 molecules-26-03042-t001:** Bioactive components of *Salvia* spp. and their antidiabetic potential.

Ref.	Objective	Methods	Findings	Conclusion
[31]	To identify the antidiabetic compounds and α-amylase inhibitors from *S. virgata*	Extraction, chromatography and spectroscopy Antidiabetic activity: α-amylase	Ethanol extract of *S. virgata* showed the α-amylase activity (IC50 = 19.08 (18.61–19.56) mg/mL) The chromatographical analysis detected flavonoid compounds (chrysoeriol) Chrysoeriol inhibited the α-amylase activity (IC50 = 1.27 (1.21–1.33) mM	Isolation and identification of flavone compound, chrysoeriol
[32]	To explore the phytochemical composition, inhibitory enzyme potential, and antioxidant activities of *S. viridis* ethanolic root extracts obtained by different extraction methods	Extraction and chemical profiling Antidiabetic activity: α-amylase and α-glucosidase	Ultrasonication-assisted extraction (UAE) extract possessed the highest phenolic and flavonoid contents (111.41 mg gallic acid and 23.46 mg rutin equivalent/g extract) α-glucosidase inhibition (1.61–1.65 mmol acarbose equivalent (ACAE)/g) and α-amylase inhibition (0.56–0.73 mmol ACAE/g)	Possible therapeutic application of *S. viridis* ethanolic root extracts for the management of chronic complications, such as diabetes
[11]	To investigate the volatile oil composition of the leaves and flowers and to evaluate the antidiabetic enzyme inhibitory assays of the oils and several extracts of *S. urmiensis*	Extraction and chemical profiling Antidiabetic activity: α-glucosidase and α-amylase	Leaves essential oil (EO) was rich in ester compounds Flowers EO was rich in 6,10,14-trimethyl-2-pentadecanone (55.7%) Antidiabetic potential (IC50 = 8–145 μg/mL) Methanol extract has the strongest α-amylase and α-glucosidase inhibitory activities (IC50 = 24 and 8.3 μg/mL, respectively)	The methanol extract exhibited remarkable antidiabetic potential
[33]	To evaluate the antidiabetic activities of *S. syriaca*	Extraction and chemical profiling of EO Antidiabetic activity: α-glucosidase and α-amylase	The major compound: spathulenol (87.4%) The most abundant phenolic components: rutin, quercetin, apigenin, rosmarinic acid (RA), and ferulic acid EO exhibited the strongest activity (IC50 = 1.18 mg/mL and 1.54 mg/mL in α-glucosidase and α-amylase assays	*S. syriaca* could be considered a valuable source of natural bioactive compounds
[34]	To evaluate the chemical composition and enzyme inhibitory activity of *S. nemorosa*	Extraction and chemical profiling Antidiabetic activity: α-glucosidase	The major compound: RA (7584 μg/g extract) The major components in EO: oxygenated sesquiterpenes Methanolic extract showed the strongest α-glucosidase inhibitory activity (IC50 = 19 mg/mL)	*S. nemorosa* may be useful for novel applications in functional food and pharmaceutical industries
[35]	To evaluate simultaneous determination and quantification of ursolic acid and oleanolic acid in 14 *Salvia* sp. and to reveal antidiabetic activities (α-glucosidase) along with correlations between antidiabetic activities and both acid contents	Extraction and compound separation Antidiabetic activity: α-glucosidase	Ursolic acid (0.21–9.76 mg/g) and oleanolic acid (0.20–12.7 mg/g) α-Glucosidase activity (IC50 = 17.6 to 173 µg/mL) The best α-glucosidase inhibitory activities with the lowest IC50 values, 17.6 and 25.9 μg/mL: *S. aucheri var. aucheri* and *S. adenocaulon* The two lowest active samples with the IC50 values of 162 μg/mL and 173 μg/mL: *S. sclarea* and *S. cilicica*	A strong correlation between both ursolic and oleanolic contents of extracts and inhibition effects on α-glucosidase activity was detected Anatolian *Salvia* sp. have great potential as functional plants in the management of diabetes
[36]	To evaluate the effect of SalA on diabetic vascular endothelial dysfunction (VED)	STZ-induced diabetic rats were treated with SalA (1 mg/kg, 90% purity) orally for 10 weeks after modeling and were given a high-fat diet (HFD) Serum indications, contractile and relaxant responses of aorta rings	Reduction in serum malondialdehyde (MDA), aortic advanced glycation end products (AGEs), nitric oxide synthase (NOS) activity and expression of endothelial NOS protein	SalA could protect against vascular VED in diabetes
[37]	To investigate the protective effects of SalA on the peripheral nerve in diabetic rats	STZ-induced diabetic rats were treated with SalA (0.3, 1 and 3 mg/kg, ig) for 8 weeks Peripheral nerve function via paw withdrawal mechanical threshold (PWMT) and motor nerve conduction velocity (MNCV) Expression of biomarkers	Increased PWMT, MNCV, AMP-activated protein kinase (AMPK) phosphorylation, upregulated peroxisome proliferator-activated receptor-gamma coactivator-1alpha (PGC-1α), silent information regulator protein3 (Sirt3) and neuronal NOS expression, but had no influence on liver kinase B1 (LKB1)	SalA has protective effects against diabetic neuropathy
[38]	To investigate the in vivo and in vitro antidiabetic effect of SalA and the underlying mechanisms	Alloxan-induced type 1 diabetic mice and STZ-induced type 2 diabetic rats received SalA treatment Measurement of glucose consumption and mitochondrial function Determination of AMPK and protein kinase B (Akt)	Reduced fasting blood glucose (FBG), fed blood glucose and enhanced glucose uptake Improved hepatic and skeletal muscle mitochondrial function Activated AMPK phosphorylation through Ca^2+^/calmodulin-dependent protein kinase kinase β (CaMKKβ)/AMPK signaling pathway No effect on insulin secretagogue and activation of PI3K/Akt signaling pathway	SalA exhibits antidiabetic effects
[39]	To evaluate the antidiabetic effect of SalB in multiple low-dose STZ-induced diabetes in rat	STZ-induced diabetic rats were treated with SalB at doses of 20 or 40 mg/kg for 3 weeks Measurement of serum glucose, insulin level and some oxidative stress markers Oral glucose tolerance test (OGTT), histological assessment, and apoptosis determination	SalB20 and SalB40 significantly decreased serum glucose and improved OGTT Serum insulin was significantly higher in SalB20- and Sal B40-treated diabetics SalB40 treatment significantly lowered MDA, raised glutathione (GSH), catalase activity The number of pancreatic islets and their area was significantly higher in the SalB40-treated diabetic group versus diabetics	Three-week treatment of diabetic rats with SalB exhibited Antidiabetic activity
[40]	To investigate the effects of SalB on diabetes-related metabolic changes in a spontaneous model of T2DM, as well as its potential molecular mechanism	Male C57BL/KsJ-db/db mice were orally treated with SalB (50 and 100 mg/kg) or metformin (positive drug, 300 mg/kg) for 6 weeks. Measurement of glucose tolerance, insulin tolerance, fasting blood glucose, serum lipids, insulin levels, and glycogen content in muscle	SalB decreased FBG, serum insulin, reduced hepatic gluconeogenic gene expression and improved insulin intolerance in db/db mice SalB100 significantly improved glucose intolerance, increased hepatic glycolytic gene expression and muscle glycogen content and ameliorated histopathological alterations of the pancreas Increased phosphorylated (p-AMPK) protein expression in skeletal muscle and liver, glucose transporter 4 (GLUT4) and glycogen synthase protein expressions in skeletal muscle, peroxisome proliferator-activated receptor alpha (PPARα) and phosphorylated acetyl CoA carboxylase (p-ACC) protein expressions in the liver	SalB displays beneficial effects in the prevention and treatment of T2DM
[41]	To investigate the effects of SalB on glycometabolism, lipid metabolism, insulin resistance, oxidative stress, and glycogen synthesis in the T2DM rat model	HFD and STZ-induced diabetic rats were randomly divided into the model group, SalB subgroups (50, 100, and 200 mg/kg), and rosiglitazone group	SalB100 and SalB200 significantly decreased blood glucose and insulin, and increased insulin sensitivity index (ISI) SalB significantly decreased hepatic glycogen and muscle glycogen	SalB can inhibit symptoms of diabetes mellitus in rats, and these effects may partially be correlated with its ISI, glycogen synthesis and antioxidant activities
[42]	To characterize the effects of *S. fruticosa* tea (SFT) treatment and of its main phenolic constituent RA on the levels and localization of the intestinal Na^+^/glucose cotransporter-1 (SGLT1), the facilitative GLUT2 and glucagon-like peptide-1 (GLP-1)	Two models of SGLT1 induction in rats were used: through diabetes induction with STZ and through dietary carbohydrate manipulation Drinking water was replaced with SFT or RA and blood parameters, liver glycogen and the levels of different proteins in enterocytes quantified	The increase in SGLT1 localized to the enterocyte brush-border membrane (BBM) induced by STZ treatment was significantly abrogated by treatment with SFT No effects were observed on GLUT2 or GLP-1 levels by SFT SFT and RA significantly inhibited the carbohydrate-induced adaptive increase of SGLT1 in BBM	SFT and RA modulate the trafficking of SGLT1 to the BBM and may contribute to the control of plasma glucose
[43]	To investigate that eucalyptol inhibited tubular epithelial cell disjunction and tubulointerstitial fibrosis stimulated by glucose	Mice were divided into three subgroups (non-diabetic db/m– control, db/db mice and db/db+eucalyptol (10 mg/kg body weight) for 8 weeks Immunohistochemical and Masson trichrome staining of kidney tissues Determination of TGF-β1 secretion and transfection assay	Eucalyptol inhibited glucose-induced expression of the mesenchymal markers of N-cadherin and α-smooth muscle actin Enhanced induction of E-cadherin and attenuated the induction of connective tissue growth factor and collagen IV by glucose Oral administration of eucalyptol blunted hyperglycemia and proteinuria, reversed tissue levels of E-cadherin, N-cadherin and P-cadherin and the collagen fiber deposition in diabetic kidneys. Furthermore, attenuated the induction of Snail1, β-catenin and integrin-linked kinase 1 (ILK1) in glucose-exposed tubular cells and diabetic kidneys, reversely enhanced glycogen synthase kinase (GSK)-3β expression	Eucalyptol may antagonize hyperglycemia-induced tubular epithelial derangement and tubulointerstitial fibrosis through blocking ILK1-dependent transcriptional interaction of Snail1/β-catenin
[44]	To investigate the phytochemical composition of *S. sclarea* EO (SSEO) and to explore their acute and subchronic antidiabetic potentials	SSEO chemical profiling Acute and subchronic antidiabetic potentials using male Swiss Webster mice	*S. sclarea* Beirut (SS-Bt) is characterized by high linalool concentration (average 40.2%) *S. sclarea* Taanayel (SS-Tl) is characterized by high linalyl acetate concentration (average 50.4%). Chemotypes 1 (owned to high linalool content), present at low altitude places of Lebanon and Poland, has shown significantly higher acute and subchronic antidiabetic activities than that of chemotype 5 (owned to high linalyl acetate content)	SSEOs have shown potential antidiabetic activities, and their EOs might be used in the future as complementary or alternative medicine in the management of diabetes and related complications
[45]	To examine the effects of lithospermic acid B (LAB) on the prevention of diabetic retinopathy in Otsuka Long-Evans Tokushima fatty (OLETF) rats, an animal model of T2DM	LAB (10 or 20 mg/kg) or normal saline were given orally once daily to 24-week-old male OLETF rats for 52 weeks Assessment of fundoscopic findings and vascular endothelial growth factor (VEGF) expression Measurement of glucose metabolism, serum levels of high-sensitivity C-reactive protein (hs-CRP), monocyte chemotactic protein-1 (MCP1), tumor necrosis factor-alpha (TNFα) and urinary 8-hydroxy-2′-deoxyguanosine (8-OHdG) levels	LAB treatment improved insulin resistance and glucose intolerance; reduced serum s-CRP, MCP1, TNFα, and urinary 8-OHdG levels; and prevented vascular leakage and basement membrane thickening	Treatment with LAB had a preventive effect on developing diabetic retinopathy
[46]	To investigate the effect of cryptotanshinone on myocardial fibrosis in diabetic rats	Male Wistar rats were separated into three groups (control, vehicle-treated STZ-treated rats, and cryptotanshinone-treated STZ-treated rats)	In STZ-treated rats, FBG levels and heart weight/body weight ratio were markedly increased, but both were not modified by cryptotanshinone Cardiac performance in catheterized STZ-treated rats was improved. The histological results from Masson staining showed that cryptotanshinone attenuated cardiac fibrosis in STZ-treated rats Both the messenger ribonucleic acid (mRNA) and protein levels of the signal transducer and activator of transcription 3 (STAT3), matrix metalloproteinase-9, and connective tissue growth factor were reduced by cryptotanshinone in high glucose-cultured cardiomyocytes	STAT3 regulates matrix metalloproteinase-9 and connective tissue growth factor expression in diabetic rats with cardiac fibrosis, cryptotanshinone inhibited fibrosis to improve cardiac function by suppressing the STAT3 pathway
[47]	To screen and explore bioactive constituents from the root of *S. miltiorrhiza Bunge* acting on renin activity and evaluates its osteoprotective efficacy in diabetic mice	Identification of tanshinone IIA (TSIIA) STZ-induced diabetic C57BL/6 mice were injected with TSIIA (10 and 30 mg/kg)	Treatment of diabetic mice with TSIIA with both doses significantly decreased angiotensin (ANG) II level in serum (from 16.56 ± 1.70 to 10.86 ± 0.68 and 9.14 ± 1.31 pg/mL) and reduced ANG II expression in bone Improved trabecular bone mineral density and microstructure of proximal tibial end, increased trabecular bone area of distal femoral end in diabetic mice	TSIIA has beneficial effects on bone of diabetic mice and potentially suggested applying *S. miltiorrhiza* in the treatment of osteoporosis and drug development of TSIIA as a renin inhibitor
[48]	To investigate the potential therapeutic function of TSIIA on diabetic cardiomyopathy in an experimental diabetic rat model	STZ-induced diabetic rats were IP injected with TSIIA for 6 weeks	Blood glucose concentration was slightly reduced in the low-dose TSIIA treatment group TSIIA injection improved cardiac function and restored the loss of mitochondrial cristae, swollen mitochondrial matrix and disorganized myofibrils in myocardial cells TSIIA injection increased the activity of superoxide dismutase and suppressed the endoplasmic reticulum (ER) stress signaling in STZ-induced diabetic rats	TSIIA may ameliorate diabetic cardiomyopathy in diabetic rats, possibly via suppressing oxidative stress and ER stress activation
[49]	To evaluate the therapeutic potential of the polyphenolic acids fraction (PAF) from *S. miltiorrhiza* Bunge in the T2DM rats model	Four groups of rats were orally administrated with an oral dose of 187 mg/kg PAF for 28 days	PAF induced a significant decrease in FBG, fasting blood insulin (FINS), total cholesterol (TC), triglyceride (TG) and blood urea nitrogen (BUN), and increased ISI in diabetic rats induced by an HFD and a low dose of STZ	PAF is an effective fraction with antidiabetic potential
[50]	To investigate the effects and possible mechanisms of the pharmacodynamic interaction between paeonol (Pae) and danshensu (DSS) on cerebrovascular malfunctioning in diabetes	Diabetic rats were treated with Pae, DSS, and Pae + DSS for 8 weeks Pae, DSS, and Pae + DSS effects on vessel relaxation with or without endothelium as well as on the basal tonus of vessels from normal and diabetic rats Oxidative stress indexes	The cerebral arteries from diabetic rats show decreased vascular reactivity to acetylcholine (ACh), which was corrected in Pae, DSS, and Pae + DSS treated groups Antagonized relaxation responses increased in DSS and Pae + DSS-treated diabetic groups compared with those in diabetic and Pae-treated diabetic groups Superoxide dismutase activity and thiobarbituric acid reactive substances content significantly changed in the presence of Pae + DSS	Both Pae and DSS treatments prevent diabetes-induced vascular damage. The combination of Pae and DSS produced significant protective effects through the reduction of oxidative stress and intracellular Ca^2+^ regulatory mechanisms
[51]	To investigate the role of AGE-mediated neuroinflammation in learning and memory deficits and the effect of DSS on the cognitive decline in diabetic mice	C57BL/6 mice were injected intraperitoneally with STZ Sodium salt of DSS was administered at a dose of 15, 30, or 60 mg/kg for 12 weeks	DSS reduced the mean escape latency and increased the percentage of time spent in the target quadrant DSS partly blocked the expression of receptor of glycation end (RAGE), p-p38, and cyclooxygenase 2 (COX-2), and nuclear factor kappa-light-chain-enhancer of B cells (NF-κB) activation, and inhibited the increase of TNF-α, interleukin 6 (IL-6), and prostaglandin E_2_ (PGE₂)	DSS may provide a potential alternative for the prevention of cognitive impairment associated with diabetes by attenuating AGE-mediated neuroinflammation
[52]	To examine the protective efficacy of Jiangtang decoction (JTD) in diabetic nephropathy (DN) and elucidate the underlying molecular mechanisms	KK-Ay mice diabetic model administered JTD (12 weeks) Assessment of renal function JTD protective renal effect via pathological staining Determination of inflammatory biomarkers Analysis of phosphoinositide 3-kinase (PI3K)/Akt signaling pathway and NF-κB	A significant amelioration in glucose and lipid metabolism dysfunction. JTD treatment reduced the accumulation of AGEs and RAGE, upregulated IRS-1, and increased the phosphorylation of both PI3K (p85) and Akt JTD administration reduced the elevated levels of renal inflammatory mediators and decreased the phosphorylation of NF-κB p65	JTD might reduce inflammation in DN through the PI3K/Akt and NF-κB signaling pathways
[53]	To evaluate the effect and potential mechanism of Huangqi-danshen decoction (HDD) in the treatment of DN in a T2DM animal model, db/db mice	HDD extract was administered orally to db/db mice at a dose of 6.8 g/kg/day for 12 weeks Biochemical and pathological examinations	HDD substantially reduced urinary albumin excretion and improved renal injury in db/db mice HDD treatment significantly reversed the enhanced mitochondrial fission and PTEN-induced putative kinase 1 (PINK1)/Parkin-mediated mitophagy in the db/db mice	HDD could protect against T2DM-induced kidney injury, possibly by inhibiting PINK1/Parkin-mediated mitophagy
[54]	To assess the effectiveness and safety of compound danshen dripping pill (CDDP) in treating patients with nonproliferative diabetic retinopathy (NPDR)	223 NPDR patients were enrolled Subjects received oral study medications three times daily for 24 weeks The four groups were placebo, low-dose (270 mg), mid-dose (540 mg) and high dose (810 mg herbal medicine) Primary endpoints were changes in fluorescence fundus angiography (FFA) and fundoscopic examination parameters	For the FFA, the high-dose and mid-dose CDDP groups were 74% and 77%, respectively, significantly higher than 28% in the placebo group For fundoscopic examination, the high-dose and mid-dose CDDP groups were 42% and 59%, respectively, significantly higher than 11% in the placebo group No serious adverse events were observed	Data demonstrated the therapeutic value and safety of a danshen-containing Chinese herbal medicine in patients with diabetic retinopathy
[55]	To determine the effectiveness and safety of Tricardin along with physical rehabilitation in diabetic patients with polyneuropathies and to observe the effect of this drug on the quality of life of these patients	A randomized controlled study on 100 diabetic patients with established polyneuropathies (n = 50 intervention and 50 control) In the control group, usual therapy for diabetes plus physical rehabilitation comprising of foot care education, pain management using transcutaneous electrical nerve stimulation (TENS), nerve gliding exercises for lower limbs, balance and proprioception training was provided Tricardin capsule was administered three times daily for eight weeks (experimental group) Pre and post-treatment quality of life via self-structured questionnaire	The pretreatment quality of life score was lower in both groups, with 29.54 ± 1.85 (control) and 34.82 ± 4.78 (treatment) Quality of life score improved significantly in the intervention group (134.2 ± 9.74) (*p* = 0.01), while only slight change with control (69.08 ± 4.91)	Tricardin was found to be a safe and effective treatment option for the management of diabetic polyneuropathies
[56]	To investigate the involvement of collagen IV (ColIV) and fibronectin (FN) in the occurrence and development of DN and the effects of telmisartan and *S. miltiorrhiza* injection in the treatment of the patients	Two hundred and fifty-eight patients with stage IV DN (cases group) and 110 normal healthy subjects (control) Involved patients were subdivided into different groups according to different treatment therapies: (T group: oral telmisartan), (S + T group: *S. miltiorrhiza* injection + telmisartan) and placebo group	Glycemic and renal damage indexes indicated trends downwards both in the T group and the S + T group S + T group levels were much lower than the T group (all *p* < 0.05) Fasting blood glucose, 2 h postprandial glucose (2hPPG), glycosylated hemoglobin (HbA1c), blood urea nitrogen, serum creatinine and urinary albumin excretion rate were significant after treatment (all *p* < 0.05) Co1IV and FN in the urine were increased before intervention in the case group compared to the control (all *p* < 0.05) There were remarkable differences in Co1IV and FN levels in the urine when compared among three different intervention groups after treatment (*p* < 0.05)	*S. miltiorrhiza* injection with telmisartan has beneficial synergistic effects for DN patients through attenuating the increase in ColIV and FN, reversing hyperglycemia state and postponing ultrastructure changes of the glomerular basement membrane
[57]	To evaluate the efficacy and safety of *S. officinalis* combined with statin in dyslipidemic T2DM	Determination of the total flavonoid, total phenolic and quercetin contents The effects of 2-month extract intake (500 mg capsule three times a day) as an add-on to daily use of 15 mg glyburide, 2000 mg metformin and 10 mg atorvastatin on the blood levels of fasting blood sugar (FBS), 2hPPG, HbA1c, TC, low-density lipoprotein cholesterol (LDL), TG, high-density lipoprotein (HDL), serum aspartate aminotransferase (AST), serum alanine aminotransferase (ALT), creatinine and body mass index were studied in 50 patients and compared with the placebo group (*n* = 50)	Total flavonoid, phenolic, and quercetin contents were 39.76 ± 3.58 mg of rutin equivalents (mean ± SD), 30.33 ± 1.23 mg of gallic acid (mean ± SD) and 0.13 mg, respectively The extract lowered FG, 2hPPG, HbA1c, TC, LDL and TG levels but increased HDL level compared to the placebo (*p* < 0.05)	The addition of the extract to statin therapy is safe and further improves lipid profile; however, conduction of more clinical trials are warranted

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
