# Peer review of "Bioactive Components of Salvia and Their Potential Antidiabetic Properties: A Review"

_molecules, 2021, doi:10.3390/molecules26103042_

Round 1

Reviewer 1 Report

Q: Pgae1-Line23:  Substitute the word “probable” with “possible”

Q: Pgae1-Line36:  Use appropriate word instead of “Lately”

Q: Page 2-lines 42-45: the sentences were not in a meaningful way and hence, recommended to revise

     in a good manner

Q: Page 2-lines 47-48: the sentences were repeated in meaning! and not in good verbal form.

Q: Page 2-line 61: Substitute the word “Past” with “Earlier”

Q: Page 2-lines 66&71: The sentences should start with capital letter i.e.  “ a-Glucosidase”

Q: page 2-lines 66-80: I recommend revising these lines in a systematic manner with meaningful words. For ex: The importance role of a-glucosidase and the complications of usage of inhibitors of a-glucosidase in the treatment of T2DM.

Q:  I recommend the authors concise the data in table -1 with precise information. Ex:  Objective: a-amylase inhibitors or a-Glucosidase inhibitors; Method: in vitro/ in vivo(cell lines)-MeOH extract; Findings: EtOH extract showed potent inhibition of a-amylase (IC50= 19.08 mg/mL) ; Conclusion: isolates metabolites/ best metabolite/extract etc. 

Q: Page 22-line 151-152: There was no verbal form. For-ex: The study is aimed to investigate the association of possible disease mechanisms with their chemical composition of species or definite plant extract of species.

Q: Page 22-line 162-170: These prescribed sentences are of no significance.

Q:  I recommend the authors please summarize accurate and significant information under the section Discussion.  For instance: Are there any differences in activities of the same metabolite in different species? The same chemical compound has different activity profiles with different enzymes of diabetes. What species is a rich source of secondary metabolites or anti-diabetic compounds etc.

Q:  I recommend the authors describe the conclusions with accurate outcomes. For ex. which metabolite? What plant extract? Which particular species of Salvia are potent to treat diabetes? What type of isolated compounds are potent inhibitors of multiple targets of diabetes.

Q: The author selected significant work but it is out of logic. I recommend the authors revise the manuscript in a concise manner with attentive results. Several sentences were meaningless and non-verbal form. Hence, please revise the manuscript with the subject expert as well as a negative English speaker.

Author Response

Dear Editor,
The manuscript has been revised according to the suggestions and comments of the reviewers. 

Q1: Page1-Line23:  Substitute the word “probable” with “possible”

Response 1: The correction has been made

Q2: Page1-Line36:  Use appropriate word instead of “Lately”

Response 2:  The correction has been made

Q3: Page 2-lines 42-45: the sentences were not in a meaningful way and hence, recommended to revise in a good manner

Response 3: We have revised the sentence in a meaningful way.

Q4: Page 2-lines 47-48: the sentences were repeated in meaning! and not in good verbal form.

Response 4: The repetition has been removed

Q5: Page 2-line 61: Substitute the word “Past” with “Earlier”

Response 5: The correction has been made

Q6: Page 2-lines 66&71: The sentences should start with capital letter i.e.  “ a-Glucosidase”

Response 6: The correction has been made

Q7: page 2-lines 66-80: I recommend revising these lines in a systematic manner with meaningful words. For ex: The importance role of a-glucosidase and the complications of usage of inhibitors of a-glucosidase in the treatment of T2DM.

Response 7: The sentences have been revised accordingly

Q8:  I recommend the authors concise the data in table -1 with precise information. Ex:  Objective: a-amylase inhibitors or a-Glucosidase inhibitors; Methodin vitroin vivo(cell lines)-MeOH extract; Findings: EtOH extract showed potent inhibition of a-amylase (IC50= 19.08 mg/mL) ; Conclusion: isolates metabolites/ best metabolite/extract etc. 

Response 8: The table is modified/corrected as recommended by both reviewers

Q9: Page 22-line 151-152: There was no verbal form. For-ex: The study is aimed to investigate the association of possible disease mechanisms with their chemical composition of species or definite plant extract of species.

Response 9: This statement is included to give a brief introduction of the topic and does not specify any study. Thus, we feel that this sentence is to be included with slight modification in a meaningful way.

Q10 : Page 22-line 162-170: These prescribed sentences are of no significance.

Response 10: This statement is included to mention the multiple techniques available for characterisation studies. Hence, we retain this sentence in the current form. 

Q 11:  I recommend the authors please summarize accurate and significant information under the section Discussion.  For instance: Are there any differences in activities of the same metabolite in different species? The same chemical compound has different activity profiles with different enzymes of diabetes. What species is a rich source of secondary metabolites or anti-diabetic compounds etc.

Response 11: The discussion section is revised to correlate with the findings in the Table. Significant information that indicates the importance of the study was added and emphasised were necessary.

Q12:  I recommend the authors describe the conclusions with accurate outcomes. For ex. which metabolite? What plant extract? Which particular species of Salvia are potent to treat diabetes? What type of isolated compounds are potent inhibitors of multiple targets of diabetes.

Response 12: The conclusion is revised in the best possible form to highlight the significance of the review. 

Q13: The author selected significant work but it is out of logic. I recommend the authors revise the manuscript in a concise manner with attentive results. Several sentences were meaningless and non-verbal form. Hence, please revise the manuscript with the subject expert as well as a negative English speaker

Response 13: We have revised the manuscript accordingly to make it more readable, engaging and clearer.

Reviewer 2 Report

This review manuscript collects the recent studies on the bioactive components of Salvia and their antidiabetic mechanism. Comments are as follows: 

There is a room to improve the table summary. It should be revised more concisely. The table list numbers are not linked to the explanation in the discussion.   

Uppercase letters at the beginning of the sentence. For example, alfa-glucosidase => alfa-Glucosidase  

Author Response

Dear Editor,
The manuscript has been revised according to the suggestions and comments of the reviewers. 

Q1: There is a room to improve the table summary. It should be revised more concisely. The table list numbers are not linked to the explanation in the discussion.  

Response 1: The table is modified/corrected as recommended by both reviewers

Q2: Uppercase letters at the beginning of the sentence. For example, alfa-glucosidase => alfa-Glucosidase  

Response 2: The correction has been made accordingly.

Round 2

Reviewer 1 Report

Q. I recommend the authors revise table -1 with more precise information about the past findings. 

Q. A major improvement is needed in the discussion section for the association of possible disease mechanisms with their chemical composition of plant species. 

Q. I could not find a major change in the terminology in the description of the manuscript in its revision.

Author Response

We would like to thank the reviewer for their time, effort and detailed comments. The revision made is written in red.

Q. I recommend the authors revise table -1 with more precise information about the past findings. 

Response 1:  The table has been revised to make it more readable and easy to understand.

Q. A major improvement is needed in the discussion section for the association of possible disease mechanisms with their chemical composition of plant species. 

Response 2: We have made some improvements in the discussion. However, the mechanism of action of chemical compounds appears not to be dictated to any great degree and remained elusive.

Q. I could not find a major change in the terminology in the description of the manuscript in its revision.

Response 3: We have done the necessary changes.

Reviewer 2 Report

The manuscript is adequately revised. I recommend this review manuscript for publication to Molecule.  

Author Response

We would like to thank you the reviewer for accepting our paper. Thank you for your time, effort and detailed comments.